# S$^2$R$^2$: Semantic Segment Robustness Regularisation on Prompt Perturbation

## Abstract

Large language models (LLMs) are highly sensitive to prompt perturbations, where small changes to key segments can lead to unreliable outputs. Existing robustness methods often optimise holistic objectives, overlooking semantic asymmetry and lacking certified guarantees. In this work, we propose Semantic Segment Robustness Regularisation (S$^2$R$^2$), a fine-tuning framework based on Low-Rank Adaptation (LoRA) that enforces segment-level alignment and penalises perturbation-induced attention shifts. We demonstrate that this objective is connected to a Probably Approximately Correct (PAC)-Bayesian generalisation bound, which can be formally tightened by constraining the LoRA parameter norms. Experiments across multiple models and domains show that S$^2$R$^2$ consistently reduces empirical risk, achieves significantly tighter bounds than strong baselines, and transfers effectively to out-of-distribution data.

## 1 Introduction

Large Language Models (LLMs) have achieved widespread adoption in numerous applications. However, their reliability is often compromised by minor imperceptible perturbations to input prompts, which can lead to unreliable or even malicious outputs (Hu et al., 2024; Honarvar et al., 2025; Wang et al., 2024; Zhu et al., 2024b). For example, a summarisation model may generate an incorrect medical conclusion if one clinical term is misspelt. This fragility undermines the utility of LLMs and poses significant risks in safety-critical domains. Therefore, robustness is a prerequisite for the trustworthy development of LLMs (Xhonneux et al., 2024; Tao et al., 2024; Paulus et al., 2025).

Many researchers have focused on bolstering LLM robustness (Lin et al., 2025; Gan et al., 2024; Rauba et al., 2024; Marjanovic et al., 2024; Wang et al., 2023) and provided well-designed fine-tuning strategies (Qiang et al., 2024; Wu et al., 2021; Aghajanyan et al., 2021; Zhu et al., 2020; Jiang et al., 2020). A common thread in these methods is a "holistic" treatment of the output, for example, by minimising the Kullback-Leibler (KL) divergence over an entire sequence. Yet, this approach disregards a core principle of language: Semantic information is unevenly distributed in a sentence. Just as a few keywords can define a sentence's message, particular segments of an LLM output are more critical to its semantic integrity. This principle of non-uniform impact is also seen in studies on adversarial fairness (Agarwal et al., 2018; Hashimoto et al., 2018; Jin et al., 2025). By ignoring this, holistic methods fail to account for the unbalanced vulnerability of text, where damage to key semantic segments can be disproportionately harmful (Qiang et al., 2024).

Focusing on semantic asymmetry is important, but it only captures part of the picture. The internal reasoning dynamics of the model rely heavily on the attention mechanism (Vaswani et al., 2017). Perturbations on model input influence output performances (Gan et al., 2024; Agrawal et al., 2025) by inducing shifts into both embeddings and attention score matrices. However, existing research on robustness does not deeply investigate the influence of the attention mechanism itself. Instead, these studies limit their scope to empirically aligning the outputs from perturbed inputs with those from the clean. Furthermore, such empirical methods also provide no certified guarantees of robustness on unseen data, leaving open questions about their generalisation ability.

Therefore, in this research, we introduce Semantic Segment Robustness Regularisation (S$^2$R$^2$), a new fine-tuning framework based on Low-Rank Adaptation (LoRA) (Hu et al., 2022), designed to

Figure 1: Overview of $S^2R^2$. A clean input and its perturbed variant are processed through a LoRA-based fine-tuned LLM. $S^2R^2$ minimises two complementary objectives: **(1)** Output-oriented segment-level semantic loss $L_{sem}$ penalises worst-case semantic shifts. **(2)** Mechanism-oriented attention shift loss $L_{att}$ constrains perturbation-induced changes in LoRA parameters. Together with the base cross-entropy loss $L_{CE}$, these objectives tighten the two terms of the PAC-Bayesian generalisation bound. We also established the connection between $L_{att}$ and regularisation.

address the gaps above. We move beyond the purely empirical objectives towards certified robustness, and meanwhile consider the asymmetry of semantic information. LoRA provides a tractable hypothesis space for the Probably Approximately Correct (PAC) Bayesian framework (McAllester, 1999) from its parameter-efficient nature, which is central to our theoretical analysis. $S^2R^2$ operates as shown in Fig. 1. First, instead of holistic output comparisons, it strategically focuses on the worst-case semantic segments. Second, it introduces a regulariser that directly penalises attention shifts, promoting a more stable internal reasoning process under perturbation. To ensure these empirical improvements are not an artefact of overfitting, we derive a formal guarantee on the model's generalisation performance according to the PAC-Bayesian bound, bridging a critical gap between empirical findings and theoretical assurance.

To summarise, the main contributions of our paper are as follows:

**C1:** We formalise segment-level robustness and propose a targeted mechanism to protect key parts of the output, moving beyond simplistic token or sentence-level comparisons for transformer-based architectures. **C2:** We introduce an explainable regulariser based on cross-attention shifts that can serve as a training objective. It not only improves robustness but also constrains parameter updates to enhance generalisation. **C3:** We derive a closed-form PAC-Bayesian bound for robust LoRA fine-tuning, providing the first certified generalisation guarantee for LLMs fine-tuned against prompt perturbations[1].

## 2 PRELIMINARIES

To bridge the gaps highlighted in Sec. 1, our work is guided by three questions:

**Q1:** How do input perturbations during fine-tuning affect the model's internal reasoning process?

**Q2:** How to improve the performance of the robustness of the fine-tuning process?

**Q3:** Can this robust fine-tuning approach guarantee generalisation and find an existing formal generalisation risk upper bound?

To address these questions, we first review existing literature to identify the gaps.

### 2.1 ROBUSTNESS TO INPUT PERTURBATION

LLMs often exhibit sensitivity to minor, semantically preserving perturbations in the input (Wang et al., 2024; Agrawal et al., 2025), ranging from unintentional typos (Gan et al., 2024; Dong et al.,

---

[1]For transparency, we note that an LLM was used to assist with language polishing. See detailed statement in App. F. The source code for this paper will be made publicly available upon acceptance.

2023) and paraphrasing (Wang et al., 2023) to deliberate adversarial attacks. To optimise LLMs' performances accordingly, several robustness fine-tuning strategies have been developed. Data augmentation enriches the training set with perturbed examples to expose the model to a wider variety of inputs (Wei & Zou, 2019), as well as more sophisticated methods such as back-translation (Edunov et al., 2018). Adversarial training generates worst-case examples to maximise the training loss compared with the ground truth. Since text is discrete, projected gradient descent has been used in the continuous embedding space (Waghela et al., 2024). Consistency-based methods add a penalty term between a model's outputs for a clean input $x$ and a perturbed one $x'$ to encourage smoother model behaviour, measured by the KL divergence (Aghajanyan et al., 2021) or Jensen-Shannon (JS) divergence (Qiang et al., 2024) between their respective output probability distributions. (Jiang et al., 2020) and (Zhu et al., 2020) merge the consistency into adversarial input production to achieve the state-of-the-art (SOTA) performances. Although current approaches are dedicated to robustness optimisation, they often holistically minimise the loss between the entire model output and the target sequence. This overlooks the underlying mechanisms of how perturbations induce uncertainty. Returning to traditional deep learning, it is highlighted that not all components contribute equally to the robustness (Xu et al., 2021; Jin et al., 2025). Drawing from this, we examine the spectral influences from the perspective of language.

## 2.2 LLMs FINE-TUNING

Parameter-Efficient Fine-Tuning (PEFT) approaches use prompt-based (Lester et al., 2021; Liu et al., 2024b) and adapter-based (Hu et al., 2022; Liu et al., 2024a; Dettmers et al., 2023; Jiang et al., 2024) methods to circumvent the prohibitive cost of updating and storing every parameter in a large model. Unlike prompt-based methods, adapter-based approaches such as LoRA and its variants are built on the principle that weight updates lie in a low-rank subspace, allowing them to reduce trainable parameters while directly influencing the model's attention distributions. Given that our objective is to explore hidden representations and cross-attention behaviour, we adopt LoRA as the backbone of our method. For a Transformer layer $l$, we have:

$$\boldsymbol{W}_* = \boldsymbol{W}_{0,*} + \boldsymbol{W}_*', \text{ where } \boldsymbol{W}_*' := \boldsymbol{B}_* \boldsymbol{A}_*^\top, \boldsymbol{B}_* \in \mathbb{R}^{d_{out} \times r}, \boldsymbol{A}_* \in \mathbb{R}^{d_{in} \times r}, r \ll \min(d_{in}, d_{out}),$$

$r$ is the adapter matrix rank, the subscript $*$ is a wildcard for the transformer Query, Key, or Value matrices, the pre-trained weight matrix $\boldsymbol{W}_{0,*} \in \mathbb{R}^{d_{out} \times d_{in}}$ remains frozen over fine-tuning, $\boldsymbol{B}_*$ and $\boldsymbol{A}_*$ are trainable. Then the Query matrices for the former hidden layer $\boldsymbol{H}$ become:

$$\boldsymbol{Q} = \boldsymbol{H}(\boldsymbol{W}_{0,Q} + \boldsymbol{Q}') \text{ with } \boldsymbol{Q}' = \boldsymbol{H} \boldsymbol{B}_Q \boldsymbol{A}_Q^\top, \text{ and } \boldsymbol{K}, \boldsymbol{V} \text{ can be updated analogously.}$$

To define a prior and posterior distribution over LoRA trainable parameters, Gaussian distributions (Goodman, 1963) can mathematically formalise the belief that task adaptation requires a minimal perturbation from the pre-trained state. This practice has been well-established within the broader Bayesian deep learning literature (Blundell et al., 2015; Dziugaite & Roy, 2017). Considering the fine-tuning dataset is typically small relative to the initial pre-training corpus, it is insufficient to change the variance drastically. Therefore, we can assume:

**Assumption 1.** *The data-independent prior distribution $P = \mathcal{N}(0, \tau^2 I)$ and the data-dependent posterior $Q = \mathcal{N}(\mu, \sigma^2 I)$ can be described by Gaussian distributions with comparable variances $\tau^2$ and $\sigma^2$.*

During LoRA fine-tuning, $\boldsymbol{B}_*$ and $\boldsymbol{A}_*$ tend to remain balanced in magnitude, rather than one matrix growing disproportionately large while the other shrinks. This behaviour has been partly attributed to factors such as the symmetric gradient structure of the matrix product and the implicit regularisation of stochastic gradient descent(Gunasekar et al., 2017). Moreover, an empirical examination by (Zhu et al., 2024a) illustrates that even though $\boldsymbol{B}_*$ and $\boldsymbol{A}_*$ hold asymmetry in their data extraction responsibility, in the standard LoRA training paradigm, the magnitudes of the learned matrices $\boldsymbol{B}_*$ and $\boldsymbol{A}_*$ are often observed to be comparable. Therefore, we can assume:

**Assumption 2.** *The Frobenius Norms (computationally efficient and differentiable for each element of a matrix) of LoRA matrices $\|\boldsymbol{B}_*^l\|_F$ and $\|\boldsymbol{A}_*^l\|_F$ are comparable.*

See empirical validation in App. E.

## 2.3 PERTURBATION EFFECT

To answer **Q1**, we need to first formally characterise input perturbations' mathematical impact on the attention mechanism. When the input layer $H$ is perturbed by a small error term $\varepsilon$ ($\|\varepsilon\|_\infty \leq \epsilon$) the resulting pre-softmax attention score vector $\alpha$ can be decomposed. Let $Q_0 = H \cdot W_{Q,0}$ be the original Query matrix derived from the frozen pre-trained weights. Our proposed robust fine-tuning method introduces LoRA matrices $B'$ and $A'$, while the standard LoRA matrices remain $B$ and $A$. The full attention score vector $\alpha$ is then:

$$\alpha = (\underbrace{H \cdot W_{Q,0}}_{\text{Original}} + \underbrace{(H + \varepsilon)B'A'^\top}_{\text{Trainable}} + \underbrace{\varepsilon W_{Q,0}}_{\text{Uncontrollable}}) \frac{K^\top \cdot V}{\sqrt{d_k}}, \quad (1)$$

where the **Original** component represents the model's original baseline, pre-trained behaviour. The **Trainable** component is the low-rank update controlled during fine-tuning, producing a task-specific offset that adapts the model's behaviour to the current task. The **Uncontrollable** component represents a direct and stochastic influence of the attention logits from the perturbation, placing it outside the direct control of the trainable LoRA parameters.

Therefore, the optimisation process guides the trainable component to perform a dual function: not only to generate a *task-adaptation offset* but also to actively produce a *corrective offset* that counteracts the uncontrollable noise. The mechanistic insight is further discussed in App. B.

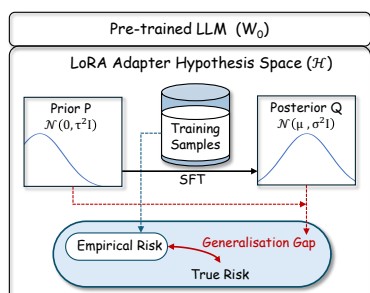

## 3 METHODOLOGY

### 3.1 PROBLEM FORMULATION

To answer **Q2** and **Q3**, we need to consider both empirical loss and generalisation ability. Therefore, we leverage the PAC-Bayesian framework (McAllester, 1999; 2003) to transform the problems into finding an upper bound of the generalised true risks given finite available training data. It is further developed and applied in neural networks (Catoni, 2007; Dziugaite & Roy, 2017) especially the Gibbs classifier (Morvant et al., 2012), and (Jin et al., 2025) discusses the "fairness" among classes during robustness training of classifiers, which is

Figure 2: The LoRA adapter PAC-Bayesian generalisation framework. It operates within the LoRA hypothesis space $\mathcal{H}$ on top of a frozen pre-trained LLM weight $W_0$. A data-independent *Prior P* represents our initial belief over the LoRA parameters. Supervised fine-tuning on training samples updates this belief to a data-dependent *Posterior Q*. The PAC-Bayesian theorem bounds the *True Risk* using the *Empirical Risk* and a complexity term, the *Generalisation Gap*, which is determined by the $D_{\text{KL}}$ between $Q$ and $P$.

similar to our purpose of mitigating output asymmetry. Applying to LLMs is made tractable by LoRA as shown in Fig. 2. Let $\mathcal{H}$ be a hypothesis space where each hypothesis model instance $h_\theta \in \mathcal{H}$ is parameterised by $\theta \in \Theta$. Given a training set of samples $z$ drawn i.i.d. from an unknown data distribution $\mathcal{D}$, the performance of a hypothesis is evaluated by a loss function $L(h, z)$. The objective of learning is to find a hypothesis with low true risk, $L_{tr}^{\mathcal{D}} := \mathbb{E}_{z \sim \mathcal{D}} L_{tr}(h_\theta, z)$, given a confidence level of $1 - \delta$ and available training samples $z \sim \mathcal{D}_t$ that leads to the empirical risk $L_{ex}^{\mathcal{D}_t} := \mathbb{E}_{z \sim \mathcal{D}_t} L_{ex}(h_\theta, z)$. The inequality between the prior and posterior distributions bounds the expected true risk by the expected empirical risk plus a complexity term, measured by the Kullback-Leibler divergence $D_{\text{KL}}$. For adversarial fine-tuning scenarios over limited training samples, we utilise a common and tight form of the PAC-Bayesian bound. This form can be derived from more general inequalities by optimising the trade-off parameter, resulting in the following expression (Seeger, 2003; Catoni, 2007):

$$L_{tr}^{\mathcal{D}} \leq \underbrace{L_{ex}^{\mathcal{D}_t}}_{\text{Empirical}} + \underbrace{\sqrt{\frac{D_{\text{KL}}(Q(\theta)\|P(\theta)) + \ln(\frac{2\sqrt{n}}{\delta})}{2n-1}}}_{\text{Complexity}}, \quad \delta \in (0,1), \quad (2)$$

where $L_{ex}^{\mathcal{D}_t}$ is the empirical loss after training on available samples, $n$ is the number of samples, and $\theta$ represents the trainable model parameters from LoRA structures as shown in Fig. 2.

The complexity term quantifies the generalisation error through including the influences of $D_{\text{KL}}$ between the posterior distribution $Q(\theta)$ and the prior $P(\theta)$, confidence, and sampling scale. This term offers several intuitive interpretations regarding the bound's behaviour:

**(1)** The confidence parameter $\delta$ embodies a trade-off between certainty and tightness. Higher confidence $(1 - \delta)$ necessarily widens the bound, reflecting higher certainty of its validity. **(2)** The $D_{\text{KL}}$ acts as a regulariser. A large divergence indicates that the posterior distribution $Q$ has moved far from the prior belief $P$ to fit the training data. The bound penalises this complexity as such a significant shift may lead to overfitting, thus warranting a looser guarantee. **(3)** The number of samples $n$ ensures that the bound tightens as more data is observed at a rate of $O(n^{-\frac{1}{2}}\sqrt{\ln n})$. This aligns with the principle that the empirical risk gradually approaches true risk as the sample size grows.

Therefore, our strategy is to minimise this bound by jointly addressing both terms, and now proceed to analyse each term individually.

## 3.2 Empirical Term

According to our analysis in Sec. 2.3, model robustness should be considered based on both external behavioural changes: altered output semantics, and internal mechanistic shifts: attention pattern change. We use the following two Shift statistics to measure the changes caused by perturbations: **semantic shift** and **attention shift**.

### 3.2.1 Semantic shift

Existing holistic consistency losses minimise divergence over the entire sequence, but this dilutes robustness signals by spreading gradients evenly across all tokens. We posit that the worst-case semantic deviation is bounded by the spectral properties of the perturbation operator and the content's semantic structure, which can be calculated by the product of *the largest singular values of perturbation and content semantic covariance*, which implies that the greatest semantic shift occurs when the perturbation's principal direction aligns with the content's principal semantic axis, highlighting the importance of semantically coherent units rather than entire sequences.

Therefore, we process model output by cutting text into semantic segments via lightweight discourse-based segmentation and alignment. Given a clean prompt $\boldsymbol{x}$ with a length of $T_x$ tokens and its perturbed variant $\boldsymbol{x}'$, let $\boldsymbol{\mathcal{S}}$ and $\boldsymbol{\mathcal{S}}'$ denote the sets of clean and perturbed semantic segments in the target outputs $\boldsymbol{y}$ and $\boldsymbol{y}'$, respectively. The model produces semantically segmented embeddings $\boldsymbol{e}_s$ and $\boldsymbol{e}_{s'}$ for $\boldsymbol{x}$ and $\boldsymbol{x}'$, respectively. We therefore propose a computationally feasible objective to align the homologous segments and measure the amount of meaning drift at the granularity of text segments. We define the following distance as the spectral semantic loss:

$$\mathcal{L}_{\text{sem}} = M(\boldsymbol{S}, \boldsymbol{S}') := \sum_{s \in \mathcal{S}} \Big\| \underbrace{\boldsymbol{e}_s}_{\text{Clean}} - \underbrace{\sum_{s' \in \mathcal{S}'} \boldsymbol{T}_{ss'}\, \boldsymbol{e}_{s'}}_{\text{Aligned perturbed}} \Big\|_2^2 \quad , \quad \boldsymbol{e}_s = \frac{1}{|s|} \sum_{t \in s} \boldsymbol{e}_t, \tag{3}$$

where the alignment matrix $\boldsymbol{T} \in [0, 1]^{|\mathcal{S}| \times |\mathcal{S}'|}$ is dynamically computed as the solution to an optimal Transport plan in the Monge-Kantorovich Problem (Villani, 2021), considering that perturbations can alter the sequence structure (e.g., reordering, inserting, or deleting segments), making a fixed one-to-one comparison brittle. $M$ treats the sets of clean and perturbed segment embeddings as two empirical distributions and finds the most efficient Transport plan between them, which allows our discrepancy metric to disregard structural noise and isolate the true semantic deviation.

### 3.2.2 Attention Shift

We begin by analysing the attention scores at a granular level. In Transformer architectures, an attention weight $a_{ij}$ represents the importance assigned by a query token at position $i$ to a key token at position $j$. The pre-softmax attention score vector for a specific key token $j$ is the collection of scores from all $T_x$ query positions: $\boldsymbol{\alpha}_j = [\alpha_{1j}, \ldots, \alpha_{ij}, \ldots, \alpha_{T_x, j}]$. According to our model definition in Eq. 1, the change in this score vector caused by a perturbation $\boldsymbol{\varepsilon}$, is given by:

$$\boldsymbol{\alpha}'_j = \left( \boldsymbol{\varepsilon} \boldsymbol{W}_{Q,0} + (\boldsymbol{H} + \boldsymbol{\varepsilon}) \boldsymbol{B}' \boldsymbol{A}'^{\top} - \boldsymbol{H} \boldsymbol{B} \boldsymbol{A}^{\top} \right) \frac{\boldsymbol{K}_j^{\top} \boldsymbol{V}_j}{\sqrt{d_k}}. \tag{4}$$

The final attention weights are obtained via the softmax function, $a_{ij} = \text{softmax}(\boldsymbol{\alpha}_j)_i$, which is non-linear. Assuming the perturbation is small, we employ a first-order Taylor expansion to linearise the change in a single attention weight, $a'_{ij}$. This change is propagated from the pre-softmax shifts $\alpha'_{kj}$ from all query positions $k \in \{1, \ldots, T_x\}$:

$$a'_{ij} \approx \sum_{k=1}^{T_x} \frac{\partial a_{ij}}{\partial \alpha_{kj}} \alpha'_{kj} = \frac{\partial a_{ij}}{\partial \alpha_{ij}} \alpha'_{ij} + \sum_{k \neq i}^{T_x} \frac{\partial a_{ij}}{\partial \alpha_{kj}} \alpha'_{kj} = a_{ij}(1 - a_{ij})\alpha'_{ij} - \sum_{k \neq i}^{T_x} a_{ij}a_{kj}\alpha'_{kj}. \tag{5}$$

Eq. 5 reveals how pre-softmax perturbations affect individual attention weights. Following the setting of Sec. 3.2.1, we define the segment-wise attention $\zeta_s$ as the average total attention directed to a segment $s$. We define the total perturbation on the attention weights as a matrix $\boldsymbol{\xi}$, where each element $\xi_{ij}$ corresponds to the change $a'_{ij}$ derived above. The perturbation corresponding to a specific segment $s$ is the submatrix $\boldsymbol{\xi}_s$. Defining the total perturbation matrix $\boldsymbol{\Xi} \in \mathbb{R}^{T_q \times T_x}$, each element of this matrix $[\boldsymbol{\Xi}]_{ij}$ is the linearised change. $T_q$ is the length of the query sequence, depending on the LLM type as discussed in App. A.1. Then $\boldsymbol{\xi}_s$ is defined as the submatrix of $\boldsymbol{\Xi}$ composed of the columns indexed by the segment $s$.

To measure the segment-wise attention shift, we derive an upper bound on its change $|\zeta'_s - \zeta_s|$ under the additive perturbation $\boldsymbol{\xi}_s$. While the precise definitions of the terms vary slightly across different attention mechanisms (see App. A.1), the final bound on the sensitivity robustly takes a unified form:

$$|\zeta'_s - \zeta_s| \leq \frac{\sqrt{T_q}}{\sqrt{|s|}} \|\boldsymbol{\xi}_s\|_F. \tag{6}$$

Here we can further expand the Eq. 6. The matrix $\Xi_s$ is composed of the individual attention changes $a'_{ij}$. As rigorously proven in App. A.2, we have: $|a'_{ij}| \leq C_{ij}^{(1)}\|\boldsymbol{\varepsilon}\|_2 + C_{ij}^{(2)}\|\boldsymbol{\varepsilon}\|_2 \cdot \|\boldsymbol{B}'\|_F\|\boldsymbol{A}'\|_F$, where $C_{ij}^{(1)} \triangleq \frac{2a_{ij}(1-a_{ij})}{\sqrt{d_k}}\|\boldsymbol{K}_j^T\boldsymbol{V}_j\|_2 \cdot \|\boldsymbol{W}_{Q,0}\|_F$, $C_{ij}^{(2)} \triangleq \frac{2a_{ij}(1-a_{ij})}{\sqrt{d_k}}\|\boldsymbol{K}_j^T\boldsymbol{V}_j\|_2$.

By substituting this per-element bound into the definition of the Frobenius norm, it directly follows that $\|\Xi_s\|_F$ is in turn bounded by a function of $\|B'\|_F\|A'\|_F$. We therefore introduce an attention loss designed to penalise this controlling factor:

$$\mathcal{L}_{\text{att}} = \lambda \cdot \|B'\|_F\|A'\|_F, \tag{7}$$

where $\lambda$ is a hyperparameter to balance this objective with the primary task loss. Here, we can augment the empirical loss of robustness fine-tuning in Eq. 2 by comprehensively considering these two losses with the traditional cross-entropy loss:

$$\mathcal{L}_{ex}^{\mathcal{D}_t} = \mathcal{L}_{\text{CE}} + \mathcal{L}_{\text{sem}} + \mathcal{L}_{\text{att}}. \tag{8}$$

## 3.3 GENERALISATION ERROR BY COMPLEXITY

To answer **Q3**, looking back at the Eq. 2, the upper bound was constrained by a complexity term to avoid overfitting. The $D_{\text{KL}}(Q(\theta)\|P(\theta))$ describes the generated distribution distance of trainable model parameters in LoRA layers based on a pre-trained LLM. For a transformer layer $l$, We define $\theta^l := vec(\boldsymbol{B}^l, \boldsymbol{A}^l)$. According to *Assumption 1*, we can compute and simplify the $D_{\text{KL}}$ for an LLM with $L$ fine-tuning participating layers as:

$$D_{\text{KL}} = \sum_{l=1}^{L} \frac{1}{2}\left[\frac{\|\mu^l\|_F^2}{(\tau^l)^2} + k^l\left(\frac{(\sigma^l)^2}{(\tau^l)^2} - 1 - \ln\frac{(\sigma^l)^2}{(\tau^l)^2}\right)\right] \approx \sum_{l=1}^{L} \frac{1}{2}\frac{\|\mu^l\|_F^2}{(\tau^l)^2} = \frac{1}{2}\sum_{l=1}^{L}\frac{\|\boldsymbol{B}^l\|_F^2 + \|\boldsymbol{A}^l\|_F^2}{(\tau^l)^2}, \tag{9}$$

where the numerator can be simply decomposed by $\underbrace{(\|\boldsymbol{B}^l\|_F - \|\boldsymbol{A}^l\|_F)^2}_{\text{Imbalance}} + 2\|\boldsymbol{B}^l\|_F\|\boldsymbol{A}^l\|_F$.

**(1) The first term** is a Norm Imbalance $(\|\boldsymbol{B}^l\|_F - \|\boldsymbol{A}^l\|_F)^2$ that penalises the dissimilarity between the LoRA matrices $\boldsymbol{B}$ and $\boldsymbol{A}$. According to the *Assumption 2*, this term remains small even in non-regularised LoRA. **(2)The second term** $\|\boldsymbol{B}^l\|_F\|\boldsymbol{A}^l\|_F$ is directly proportional to our proposed attention shift loss as $\mathcal{L}_{att}$ in Eq. 7. Conclusively, this loss also works as a **regulariser** that constrains model overfitting and enhances the generalisation ability.

Therefore, by minimising the empirical loss we design in Sec. 3.2, we can tighten the two terms of the PAC-Bayesian bound. We not only enhance the empirical robustness, but also *indirectly yet effectively constrain* the complexity term of the framework to guarantee generalisation. Now we also return to the idea by (Langford & Caruana, 2001; Langford, 2002; Dziugaite & Roy, 2017). Here, we can strictly tighten the two terms in the upper bound Eq. 2 through minimising the Eq. 8.

## 4 EXPERIMENT

**Experimental Setup** **(1) Task Selection** We adopt summarisation as our primary task since it both validates the theoretical analysis in a genuine generation setting and provides outputs with rich semantic structure, which are better suited for segment-level robustness evaluation than alternative tasks such as translation (lexically constrained) or keyphrase generation (too short). **(2) Models and Datasets** We validate $S^2R^2$ on diverse architectures: the encoder-decoder models BART-base (Lewis et al., 2020) and Flan-T5-base (Chung et al., 2024), and the decoder-only Mistral-7B-Instruct-v0.2 (Jiang et al., 2023). Our evaluation uses three summarisation benchmarks chosen to test distinct robustness aspects: CNN/Dailymail (Nallapati et al., 2016) for factual consistency (via high lexical overlap), XSum (Hasan et al., 2021) for semantic coherence (highly abstractive), and the technical PubMed (Canese & Weis, 2013) for domain-specific precision. **(3) Implementation Details** All of the experiments are conducted on GPU A100 40G. We adopt the R3F (Aghajanyan et al., 2021) and the SMART (Jiang et al., 2020) as canonical baselines according to Sec. 2.1, considering most recent robustness methods are variants of these and do not alter the principle relevant to our regularisation. All models are fine-tuned using LoRA. The base task is set up with the standard cross-entropy loss ($\mathcal{L}_{\text{CE}}$). The semantic segment discrepancy loss ($\mathcal{L}_{\text{sem}}$) is incorporated into the adversarial noise generation loop based on the holistic distance proposed by the baseline SMART, and is complemented by the external LoRA-aware attention shift ($\mathcal{L}_{\text{att}}$) which also works as the KL regulariser from our PAC-Bayesian analysis. The parameters update pseudocode is as Algorithm 1 in App. D. For our main experiments, we employ a computationally efficient strategy for segmenting model outputs based on natural language punctuation. An additional high-cost small-resource examination using an LLM segmentation method powered by a T5 model is provided in App. C, which indicates that the punctuation slicing provides similar performance and is time-efficient.

### 4.1 EVALUATION

#### 4.1.1 PERTURBATION TESTBED

To simulate common real-world text corruptions and assess model robustness, we apply three types of perturbations to the source articles in the test sets, creating three parallel evaluation branches, following (Qiang et al., 2024; Dong et al., 2023; Wang et al., 2023):

**(1) Typographical & Deletion** tests the tolerance to spelling errors and incomplete information. We swap characters within words with a probability of $p = 0.15$ and subsequently delete words from the text with a probability of $p = 0.10$. We follow recent TextAttack (Morris et al., 2020) and use random-typo noise instead of obsolete homophone swap. **(2) Synonym Replacement** evaluates the understanding of semantic equivalence despite variations in vocabulary. We replace words with their synonyms with a probability of $p = 0.15$ by the WordNet (Miller, 1995). **(3) Paraphrasing** poses a challenge to the model's deep semantic comprehension. We use a pre-trained T5 paraphrasing model (Chung et al., 2024) to rewrite the source text, generating adversarial examples that are syntactically and lexically divergent but semantically aligned with the original.

#### 4.1.2 EVALUATION METRICS

Based on the testbed above, we employ the following metrics to quantify capabilities.

**(1) Performance Drop Rate** (PDR) quantifies the relative degradation in ROUGE score (Lin, 2004) when the model is faced with perturbations (Agrawal et al., 2025). For each perturbation type, the PDR is calculated as $\text{PDR}_p = 1 - \frac{R_L(f_p)}{R_L(f_c)}$, where $R_L(f_p)$ and $R_L(f_c)$ are the ROUGE-L scores on the perturbed and clean datasets, respectively. Approaching 0 indicates superior robustness.

Table 1: Main experimental results. $|\text{PDR}|_{\text{avg}}$, $\Delta_{ed\text{-avg}}$, and $1\text{-SB}_{\text{avg}}$ are averaged over all perturbation types. E-Risk is the empirical risk, and PAC-B is the final PAC-Bayesian bound value. The best result in each category is in **bold**. Lower is better for all metrics.

| Method | $|\text{PDR}|\downarrow$ | | | | $1\text{-SB}\downarrow$ | | | | $\Delta_{ed}\downarrow$ | | | | $D_{\text{KL}}\downarrow$ | E-Risk $\downarrow$ | PAC-B $\downarrow$ |
|---|---|---|---|---|---|---|---|---|---|---|---|---|---|---|---|
| | Typo | Syno | Para | Avg | Typo | Syno | Para | Avg | Typo | Syno | Para | Avg | | | |
| **(a) Bart-base on CNN/Dailymail** | | | | | | | | | | | | | | | |
| R3F | 0.0949 | 0.0613 | 0.0872 | 0.0811 | 0.6402 | 0.4795 | 0.7507 | 0.6238 | 0.7524 | 0.6653 | 0.8597 | 0.7591 | 233.625 | 0.5831 | 0.6930 |
| SMART | 0.0008 | 0.0011 | 0.0176 | 0.0064 | 0.0611 | 0.0663 | 0.4176 | **0.1817** | 0.1378 | 0.1434 | 0.8806 | **0.3873** | 202.554 | 0.1821 | 0.2874 |
| $S^2R^2$ | 0.0006 | 0.0015 | 0.0057 | **0.0012** | 0.0717 | 0.0661 | 0.4470 | 0.1956 | 0.1674 | 0.1548 | 0.8892 | 0.4038 | **78.006** | 0.1966 | **0.2626** |
| **(b) Bart-base on XSum** | | | | | | | | | | | | | | | |
| R3F | 0.0320 | 0.0188 | 0.0548 | 0.0352 | 0.4166 | 0.2905 | 0.5898 | 0.4323 | 0.6748 | 0.5091 | 0.8985 | 0.6914 | 190.051 | 0.4185 | 0.5181 |
| SMART | 0.0108 | 0.0004 | 0.0264 | **0.0125** | 0.0566 | 0.1076 | 0.4014 | 0.2220 | 0.3196 | 0.2292 | 0.7558 | **0.4349** | 149.155 | 0.2223 | 0.3110 |
| $S^2R^2$ | 0.0346 | 0.0138 | 0.0072 | 0.0185 | 0.0506 | 0.1144 | 0.3766 | **0.2141** | 0.3912 | 0.2962 | 0.7775 | 0.4883 | **90.775** | 0.2219 | **0.2924** |
| **(c) T5-base on PubMed** | | | | | | | | | | | | | | | |
| R3F | 0.1266 | 0.0327 | 0.1043 | 0.0781 | 0.6613 | 0.3310 | 0.7105 | 0.5676 | 0.7225 | 0.3743 | 0.7716 | 0.6228 | 1377.258 | 0.5242 | 0.7874 |
| SMART | 0.0509 | 0.0060 | 0.0515 | **0.0321** | 0.6435 | 0.3737 | 0.8035 | 0.5676 | 0.6812 | 0.3944 | 0.8628 | 0.6461 | 1000.989 | 0.5560 | 0.7795 |
| $S^2R^2$ | 0.3229 | 0.1592 | 1.9640 | 0.8152 | 0.1281 | 0.1174 | 0.2873 | **0.1776** | 0.0766 | 0.0702 | 0.2116 | **0.1195** | 774.055 | 0.2355 | **0.4333** |
| **(d) Mistral-7B on PubMed** | | | | | | | | | | | | | | | |
| R3F | 0.0893 | 0.0568 | 0.2421 | 0.1300 | 0.5011 | 0.2144 | 0.8413 | 0.5189 | 0.2454 | 0.1419 | 0.7842 | **0.3902** | 565.093 | 0.4672 | 0.6365 |
| SMART | 0.0894 | 0.0588 | 0.2221 | 0.1235 | 0.5208 | 0.2119 | 0.7317 | **0.4881** | 0.2426 | 0.1376 | 0.7905 | 0.3903 | 461.931 | 0.4419 | 0.5951 |
| $S^2R^2$ | 0.0795 | 0.0553 | 0.0902 | **0.0749** | 0.5499 | 0.2674 | 0.8417 | 0.5530 | 0.3444 | 0.2472 | 0.7734 | 0.4550 | **58.106** | 0.4419 | **0.5530** |

Table 2: Zero-shot experimental result. Cross-dataset evaluation to assess the transferability of learned robustness. Bart-base are fine-tuned and evaluated on (a) different domains (general news to biomedical) and (b) different task styles (abstractive to extractive summarisation).

| Method | $|\text{PDR}|\downarrow$ | | | | $\Delta_{ed}\downarrow$ | | | | $1\text{-SB}\downarrow$ | | | | $D_{\text{KL}}\downarrow$ | E-Risk $\downarrow$ | PAC-B $\downarrow$ |
|---|---|---|---|---|---|---|---|---|---|---|---|---|---|---|---|
| | Typo | Syno | Para | Avg | Typo | Syno | Para | Avg | Typo | Syno | Para | Avg | | | |
| **(a) Fine-tuned on CNN/DailyMail tested on PubMed** | | | | | | | | | | | | | | | |
| R3F | 0.0952 | 0.0496 | 0.1164 | 0.0871 | 0.6649 | 0.4487 | 0.7442 | 0.6497 | 0.6461 | 0.5148 | 0.7881 | 0.6292 | 233.625 | 0.5770 | 0.6870 |
| SMART | 0.0138 | 0.0032 | 0.1392 | 0.0408 | 0.1921 | 0.3943 | 0.6124 | 0.6246 | 0.4204 | 0.4198 | 1.0340 | 0.3329 | 202.554 | 0.3328 | 0.4355 |
| $S^2R^2$ | 0.0012 | 0.0001 | 0.0662 | **0.0217** | 0.1014 | 0.0918 | 0.3665 | **0.4242** | 0.2594 | 0.2361 | 0.7771 | **0.1866** | 78.006 | 0.1939 | 0.2595 |
| **(b) Fine-tuned on XSum tested on CNN/DailyMail** | | | | | | | | | | | | | | | |
| R3F | 0.0453 | 0.0357 | 0.3634 | 0.0941 | 0.6197 | 0.6023 | 0.7846 | 0.8321 | 0.8195 | 0.8052 | 0.8716 | 0.6689 | 190.051 | 0.6277 | 0.7273 |
| SMART | 0.0031 | 0.0004 | 0.0366 | 0.0113 | 0.1264 | 0.1109 | 0.8114 | 0.5839 | 0.3193 | 0.2861 | 1.1460 | 0.3496 | 149.155 | 0.3392 | 0.4279 |
| $S^2R^2$ | 0.0003 | 0.0027 | 0.0132 | **0.0052** | 0.1159 | 0.1004 | 0.5995 | **0.5152** | 0.3049 | 0.2636 | 0.9771 | **0.2719** | 90.775 | 0.2695 | 0.3399 |

**(2)Output Consistency** are reference-free and directly measure the consistency of the model's behaviour. We compare the model's predictions on clean inputs ($f_c$) with those on perturbed inputs ($f_p$) by: **(a) Self-BERTScore** (SB) We use BERTScore (Zhang et al., 2020) to compute the semantic similarity between $f_c$ and $f_p$, measuring a model's *semantic stability*. **(b) Output Edit Rate** ($\Delta_{ed}$) We compute the normalised word-level Levenshtein distance (Chowdhury et al., 2013) between $f_c$ and $f_p$ to measure a model's *syntactic stability*, quantifying the degree of surface-level change in the output text induced by the perturbation.

According to the model's output performances according to the metrics above, we heuristically define the **empirical risk** in as $0.8(1\text{-SB})+0.1\text{PDR}+0.1\Delta_{ed}$, considering semantic stability is the most core indicator in our optimisation design. Other combinations are also acceptable provided the value is normalised to $[0, 1]$ in line with Eq. 2. Combining the generalisation gap calculated from the fine-tuned LoRA norms, we can calculate the bound values to validate our $S^2R^2$'s effectiveness.

## 4.2 RESULTS AND ANALYSIS

### 4.2.1 ROBUSTNESS ON STANDARD BENCHMARKS

With **Bart-base** on **CNN/Dailymail** (Tab. 1(a)), $S^2R^2$ achieves a $|\text{PDR}|_{\text{avg}}$ of just 0.0012, an 81% reduction over the strong SMART baseline. While its empirical risk (E-Risk) is comparable to SMART, $S^2R^2$ drastically reduces the $D_{\text{KL}}$ from 202.5 to 78.0. This yields a final PAC-Bayesian bound (PAC-B) of 0.2626, tightest among all methods, demonstrating that $S^2R^2$ finds a more generalisable robust solution. On the abstractive **XSum** dataset (Tab. 1(b)), $S^2R^2$ again secures the best PAC-B, driven by superior semantic stability (lower $1\text{-SB}_{\text{avg}}$) and a significantly smaller $D_{\text{KL}}$.

### 4.2.2 ROBUSTNESS IN SPECIALISED DOMAINS' METRICS

The results on **PubMed** are particularly illuminating. With **T5-base** (Tab. 1(c)), $S^2R^2$ exhibits a high $|PDR|_{avg}$. However, this is coupled with exceptionally low $\Delta_{ed,avg}$ and $1\text{-SB}_{avg}$ scores (best-in-class). This suggests that $S^2R^2$ produces outputs that are stable and semantically consistent with their clean-input counterparts, a property not fully captured by the n-gram-based ROUGE metric, which penalises valid semantic paraphrases that deviate from a single reference. By prioritising semantic self-consistency, $S^2R^2$ achieves the lowest E-Risk and a 44% tighter PAC-B than its closest competitor. This principle is further reinforced by the **Mistral-7B-Instruct** results (Tab. 1(d)). Here, even though $S^2R^2$'s E-Risk is slightly higher than the baselines, it achieves this with a $D_{KL}$ that is nearly **8 times smaller**. This efficiency in parameter usage results in the tightest certified generalisation bound. It strongly suggests that baseline methods may overfit to the perturbation patterns in the training data, leading to larger parameter norms and a weaker generalisation guarantee.

### 4.2.3 VISUALISATION OF THE REGULARISATION

Fig. 3 provides direct visual evidence for our central claim. Across all models and datasets, the LoRA parameter norms $\Sigma_l \|\boldsymbol{B}^l\|_F \|\boldsymbol{A}^l\|_F$ for $S^2R^2$ remain lower and grow more slowly, correlating with a smaller $D_{KL}$ and a tighter PAC-Bayesian bound. Combined with the low empirical risk reported in Tab. 1, we conclude that our $S^2R^2$'s robustness stems not from aggressive parameter tuning that risks overfitting, but from finding a more generalisable solution in a constrained hypothesis space.

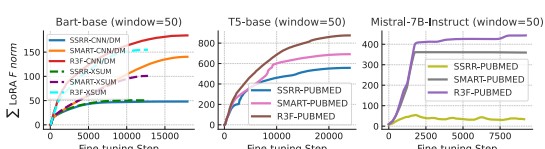

Figure 3: Evolution of the sum of Frobenius norms of LoRA matrices during fine-tuning. $S^2R^2$ consistently maintains a lower value than baselines. The x-axis differs in the three sub-figures due to the size differences of the datasets and training batches.

### 4.2.4 ZERO-SHOT CROSS-DATASET GENERALISATION

To further probe the generalisation capabilities of our framework, we conduct a challenging zero-shot cross-dataset evaluation as shown in Tab. 2. Bart-base is fine-tuned on one dataset and then directly tested against perturbations on another, unseen dataset. This setup assesses how well the learned robustness transfers across different domains and task styles. First, we assess generalisation from the general news domain to a specialised biomedical domain by training on **CNN/Dailymail** and testing on **PubMed** (Tab. 2(a)). $S^2R^2$ not only achieves the best scores **across all empirical metrics**, including a 42% reduction in E-Risk compared to the second, but also maintains the lowest KL complexity. This leads to a PAC-Bayesian bound of 0.2595, which is significantly tighter than the baselines. Next, we evaluate across different summarisation styles, training on the highly abstractive **XSum** and testing on the more extractive **CNN/Dailymail** (Tab. 2(b)). The $S^2R^2$-trained model again outperforms all baselines across all metrics.

The results suggest that $S^2R^2$ learns a more fundamental and transferable robustness mechanism, successfully avoiding overfitting to the stylistic properties of the source domain. This provides strong evidence that the generalisation guarantee offered by our framework is a meaningful predictor of real-world, out-of-distribution robustness.

## 5 CONCLUSION

This work addresses the overlooked issue of semantic asymmetry in LLM robustness, where perturbations to key segments disproportionately harm model reliability. We propose Semantic Segment Robustness Regularisation, which combines segment-level alignment with LoRA-based fine-tuning and derives a PAC-Bayesian bound for certified generalisation. Through extensive experiments on diverse summarisation tasks, we show that $S^2R^2$ achieves consistently lower empirical risk and significantly tighter bounds than strong baselines. Looking ahead, the framework can be extended to provide theoretical guarantees for other empirically driven optimisation methods.

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

# A  THEOREM PROOF SUPPLEMENTS

## A.1  SEGMENT-WISE ATTENTION SHIFT

### A.1.1  CASE 1: CAUSAL SELF-ATTENTION IN DECODER-ONLY MODELS

**Notation and Definitions.** We analyse the self-attention mechanism within a decoder-only autoregressive model. Let the model operate on a sequence of length $T_x$. We define a segment as a set of indices $s \subseteq \{1, \ldots, T_x\}$ within this sequence. The attention matrix is denoted by $A \in \mathbb{R}^{T_x \times T_x}$, where an element $a_{ij}$ represents the attention score from the query at position $i$ to the key at position $j$. For causal attention, $a_{ij} = 0$ for $j > i$. The vector of attention scores directed at a specific token $j$ from all query positions is its column vector $\boldsymbol{a}_{T_x,j} \in \mathbb{R}^{T_x}$. We then define the segment-wise attention, $\zeta_s$, as the total attention from all query positions directed to the key positions within the past segment $s$, averaged by its size $|s|$:

$$\zeta_s = \frac{1}{|s|} \sum_{j \in s} \sum_{i=1}^{T_x} a_{ij} = \frac{1}{|s|} \sum_{j \in s} \langle \boldsymbol{a}_{T_x,j}, \mathbf{1}_{T_x} \rangle. \tag{10}$$

This metric quantifies how much the model, across its entire generation process, focuses on the specific past segment $s$.

**Derivation of the Sensitivity Bound.** We introduce an additive perturbation $\boldsymbol{\xi}_j$ to each attention vector $\boldsymbol{a}_{T_x,j}$ to analyse the stability of this metric. The bound on the change, $|\zeta_s' - \zeta_s|$, is derived as follows:

$$
\begin{aligned}
|\zeta_s' - \zeta_s| &= \left| \frac{1}{|s|} \sum_{j \in s} \langle \boldsymbol{a}_{T_x,j} + \boldsymbol{\xi}_j, \mathbf{1}_{T_x} \rangle - \frac{1}{|s|} \sum_{j \in s} \langle \boldsymbol{a}_{T_x,j}, \mathbf{1}_{T_x} \rangle \right| \\
&= \frac{1}{|s|} \left| \sum_{j \in s} \langle \boldsymbol{\xi}_j, \mathbf{1}_{T_x} \rangle \right| \leq \frac{1}{|s|} \sum_{j \in s} |\langle \boldsymbol{\xi}_j, \mathbf{1}_{T_x} \rangle| \\
&\leq \frac{1}{|s|} \sum_{j \in s} \|\boldsymbol{\xi}_j\|_2 \|\mathbf{1}_{T_x}\|_2 = \frac{\sqrt{T_x}}{|s|} \sum_{j \in s} \|\boldsymbol{\xi}_j\|_2 \\
&\leq \frac{\sqrt{T_x}}{|s|} \sqrt{|s|} \left( \sum_{j \in s} \|\boldsymbol{\xi}_j\|_2^2 \right)^{1/2} = \frac{\sqrt{T_x}}{\sqrt{|s|}} \|\boldsymbol{\xi}_s\|_F.
\end{aligned}
\tag{11}
$$

where $\boldsymbol{\xi}_s$ is the matrix formed by stacking the perturbation vectors $\{\boldsymbol{\xi}_j\}_{j \in s}$.

### A.1.2 CASE 2: ENCODER SELF-ATTENTION

**Notation and Definitions.** We analyze the self-attention mechanism within a Transformer encoder that processes an input sequence of length $T_x$. We define a segment as a set of indices $s \subseteq \{1, \ldots, T_x\}$ within this **input sequence**. The attention matrix is denoted by $A \in \mathbb{R}^{T_x \times T_x}$, where an element $a_{ij}$ represents the attention score from the input token at query position $i$ to the input token at key position $j$. The vector of attention scores directed at a specific token $j$ is its column vector $\boldsymbol{a}_{T_x,j} \in \mathbb{R}^{T_x}$. The segment-wise attention, $\zeta_s$, quantifies the total attention from all input positions directed to the tokens within segment $s$, averaged by its size $|s|$:

$$
\zeta_s = \frac{1}{|s|} \sum_{j \in s} \sum_{i=1}^{T_x} a_{ij} = \frac{1}{|s|} \sum_{j \in s} \langle \boldsymbol{a}_{T_x,j}, \mathbf{1}_{T_x} \rangle.
\tag{12}
$$

**Derivation of the Sensitivity Bound.** Following an identical derivation path as in the causal self-attention case, which involves applying the triangle inequality and the Cauchy-Schwarz inequality, we arrive at the same bound on the change in $\zeta_s$ due to a perturbation $\boldsymbol{\xi}_s$:

$$
|\zeta_s' - \zeta_s| \leq \frac{\sqrt{T_x}}{\sqrt{|s|}} \|\boldsymbol{\xi}_s\|_F.
\tag{13}
$$

This bound measures how robust the model's internal representation of segment $s$ is to perturbations in attention scores.

### A.1.3 CASE 3: ENCODER-DECODER CROSS-ATTENTION

**Notation and Definitions.** We analyze the cross-attention mechanism between an encoder and a decoder. Let the encoder produce a sequence of key/value pairs of length $T_x$, and the decoder produce a sequence of queries of length $T_y$. We define a segment as a set of indices $s \subseteq \{1, \ldots, T_x\}$ within the **input (encoder) sequence**. The attention matrix is denoted by $A \in \mathbb{R}^{T_y \times T_x}$, where an element $a_{ij}$ represents the attention score from the decoder query at position $i$ to the encoder key at position $j$. The vector of attention scores directed at a specific input token $j$ is the corresponding column vector $\boldsymbol{a}_{T_x,j} \in \mathbb{R}^{T_y}$. The segment-wise attention, $\zeta_s$, quantifies the total attention from all decoder positions directed to the keys within the input segment $s$, averaged by its size $|s|$:

$$
\zeta_s = \frac{1}{|s|} \sum_{j \in s} \sum_{i=1}^{T_y} a_{ij} = \frac{1}{|s|} \sum_{j \in s} \langle \boldsymbol{a}_{T_x,j}, \mathbf{1}_{T_y} \rangle.
\tag{14}
$$

**Derivation of the Sensitivity Bound.** The derivation for the sensitivity bound follows a similar path, with the key difference being the dimension of the query space, $T_y$.

$$
|\zeta_s' - \zeta_s| = \frac{1}{|s|} \left| \sum_{j \in s} \langle \boldsymbol{\xi}_j, \mathbf{1}_{T_y} \rangle \right| \leq \frac{1}{|s|} \sum_{j \in s} |\langle \boldsymbol{\xi}_j, \mathbf{1}_{T_y} \rangle|
$$

$$
\leq \frac{1}{|s|} \sum_{j \in s} \|\boldsymbol{\xi}_j\|_2 \|\mathbf{1}_{T_y}\|_2 = \frac{\sqrt{T_y}}{|s|} \sum_{j \in s} \|\boldsymbol{\xi}_j\|_2
$$

$$
\leq \frac{\sqrt{T_y}}{|s|} \sqrt{|s|} \left( \sum_{j \in s} \|\boldsymbol{\xi}_j\|_2^2 \right)^{1/2} = \frac{\sqrt{T_y}}{\sqrt{|s|}} \|\boldsymbol{\xi}_s\|_F. \tag{15}
$$

Here, the bound is scaled by the length of the **decoder (query) sequence**, $T_y$.

## A.2 Detailed Derivations of the Attention Shift Bound

This section provides a first-principles derivation of the upper bound for the change in a single attention weight, denoted as $|a_{ij}'|$.

**Step 1: The Exact Linearised Change**  We begin with the exact first-order Taylor expansion of the change in an attention weight $a_{ij}'$ as a function of the pre-softmax score changes $a_{kj}'$. This relationship is given by:

$$
a_{ij}' = a_{ij}(1 - a_{ij})a_{ij}' - \sum_{k \neq i} a_{ij} a_{kj} a_{kj}'. \tag{16}
$$

**Step 2: Deriving a General Upper Bound via Triangle Inequality**  To derive a universally valid upper bound without making assumptions on the signs of the terms, we take the absolute value of Eq. 16 and apply the triangle inequality:

$$
|a_{ij}'| = \left| a_{ij}(1 - a_{ij})a_{ij}' - \sum_{k \neq i} a_{ij} a_{kj} a_{kj}' \right|
$$

$$
\leq \left| a_{ij}(1 - a_{ij})a_{ij}' \right| + \left| \sum_{k \neq i} a_{ij} a_{kj} a_{kj}' \right|
$$

$$
\leq a_{ij}(1 - a_{ij})|a_{ij}'| + \sum_{k \neq i} a_{ij} a_{kj} |a_{kj}'|. \tag{17}
$$

This inequality is always true. The problem now reduces to finding a uniform upper bound for the pre-softmax change components, $|a_{kj}'|$.

**Step 3: Bounding the Pre-Softmax Change Components**  The pre-softmax change vector $\boldsymbol{a}_j'$ is induced by the LoRA update. Under the assumption of a small perturbation vector $\boldsymbol{\varepsilon}$, we further assume the model has converged under an idealised robust training paradigm. In this setting, the learned structural difference between the robust model (containing $\boldsymbol{A}', \boldsymbol{B}'$) and the standard model (containing $\boldsymbol{A}, \boldsymbol{B}$) primarily serves to counteract the "expected" effect of noise. For a symmetric, zero-mean noise distribution, this expected effect is zero, implying the structural differences are themselves minimal. This allows us to posit that the dominant driver of the attention shift for a "specific" noise instance $\boldsymbol{\varepsilon}$ is the direct perturbation itself. We can therefore simplify the expression after considering the first-order effects, where the magnitude of the perturbation is represented by its norm:

$$
\boldsymbol{a}_j' \approx \frac{1}{\sqrt{d_k}}(\|\boldsymbol{\varepsilon}\|_2 \boldsymbol{W}_{Q,0} + \|\boldsymbol{\varepsilon}\|_2 \boldsymbol{B}' \boldsymbol{A}'^T)(\boldsymbol{K}_j^T \boldsymbol{V}_j). \tag{18}
$$

To establish a uniform bound for any component $|a_{kj}'| = |(\boldsymbol{a}_j')_k|$, we can use the $L_2$-norm of $\boldsymbol{a}_j'$:

$$
|a_{kj}'| \leq \|\boldsymbol{a}_j'\|_2. \tag{19}
$$

We proceed by bounding the norm $||\boldsymbol{a}_j'||_2$ using the submultiplicative property of the Frobenius norm ($||\boldsymbol{XY}||_2 \leq ||\boldsymbol{X}||_F ||\boldsymbol{Y}||_2$):

$$||\boldsymbol{a}_j'||_2 \leq \frac{1}{\sqrt{d_k}}||||\varepsilon||_2 \boldsymbol{W}_{Q,0} + ||\varepsilon||_2 \boldsymbol{B}'\boldsymbol{A}'^T||_F \cdot ||\boldsymbol{K}_j^T \boldsymbol{V}_j||_2. \tag{20}$$

Let us define this upper bound as $U_j$ for notational simplicity, such that $|a_{kj}'| \leq U_j$ for all $k$.

**Step 4: Substitution and Simplification**   We now substitute the uniform bound $U_j$ back into the inequality derived in Eq. 17:

$$|a_{ij}'| \leq a_{ij}(1 - a_{ij})U_j + \sum_{k \neq i} a_{ij}a_{kj}U_j$$

$$= U_j \left( a_{ij}(1 - a_{ij}) + a_{ij}\sum_{k \neq i} a_{kj} \right). \tag{21}$$

Given that the attention weights are the output of a softmax function, we have $\sum_k a_{kj} = 1$, which implies $\sum_{k \neq i} a_{kj} = 1 - a_{ij}$. Substituting this yields:

$$|a_{ij}'| \leq U_j \left( a_{ij}(1 - a_{ij}) + a_{ij}(1 - a_{ij}) \right)$$
$$= 2a_{ij}(1 - a_{ij})U_j. \tag{22}$$

Replacing $U_j$ with its full expression from Eq. 20, we get:

$$|a_{ij}'| \leq \frac{2a_{ij}(1 - a_{ij})}{\sqrt{d_k}}||\boldsymbol{K}_j^T \boldsymbol{V}_j||_2 \cdot ||||\varepsilon||_2 \boldsymbol{W}_{Q,0} + ||\varepsilon||_2 \boldsymbol{B}'\boldsymbol{A}'^T||_F. \tag{23}$$

**Step 5: Isolating the Norms of Trainable Matrices**   The final step is to isolate the contribution of the trainable LoRA matrices $\boldsymbol{A}'$ and $\boldsymbol{B}'$. We focus on the term $||||\varepsilon||_2 \boldsymbol{W}_{Q,0} + ||\varepsilon||_2 \boldsymbol{B}'\boldsymbol{A}'^T||_F$ and apply the triangle inequality followed by the submultiplicative property of the Frobenius norm:

$$||||\varepsilon||_2 \boldsymbol{W}_{Q,0} + ||\varepsilon||_2 \boldsymbol{B}'\boldsymbol{A}'^T||_F \leq ||||\varepsilon||_2 \boldsymbol{W}_{Q,0}||_F + ||||\varepsilon||_2 \boldsymbol{B}'\boldsymbol{A}'^T||_F$$
$$= ||\varepsilon||_2 \cdot ||\boldsymbol{W}_{Q,0}||_F + ||\varepsilon||_2 \cdot ||\boldsymbol{B}'\boldsymbol{A}'^T||_F$$
$$\leq ||\varepsilon||_2 \cdot ||\boldsymbol{W}_{Q,0}||_F + ||\varepsilon||_2 \cdot ||\boldsymbol{B}'||_F ||\boldsymbol{A}'||_F. \tag{24}$$

Substituting this result back into our main inequality gives the final bound:

$$|a_{ij}'| \leq \frac{2a_{ij}(1 - a_{ij})}{\sqrt{d_k}}||\boldsymbol{K}_j^T \boldsymbol{V}_j||_2 \left( ||\varepsilon||_2 \cdot ||\boldsymbol{W}_{Q,0}||_F + ||\varepsilon||_2 \cdot ||\boldsymbol{B}'||_F ||\boldsymbol{A}'||_F \right). \tag{25}$$

This can be expressed more concisely by defining input-dependent constants:

$$|a_{ij}'| \leq C_{ij}^{(1)}||\varepsilon||_2 + C_{ij}^{(2)}||\varepsilon||_2 \cdot ||\boldsymbol{B}'||_F ||\boldsymbol{A}'||_F, \tag{26}$$

where

$$C_{ij}^{(1)} \triangleq \frac{2a_{ij}(1 - a_{ij})}{\sqrt{d_k}}||\boldsymbol{K}_j^T \boldsymbol{V}_j||_2 \cdot ||\boldsymbol{W}_{Q,0}||_F,$$

$$C_{ij}^{(2)} \triangleq \frac{2a_{ij}(1 - a_{ij})}{\sqrt{d_k}}||\boldsymbol{K}_j^T \boldsymbol{V}_j||_2.$$

This final expression rigorously demonstrates that the change in a single attention weight is upper-bounded by a term directly proportional to the product of the Frobenius norms of the trainable matrices $\boldsymbol{A}'$ and $\boldsymbol{B}'$, providing a direct theoretical justification for regularisation strategies targeting these norms.

## B  EQUATION DISCUSSION

In this section, we will further discuss the equations shown in the main text.

For Eq. 1:

$$\boldsymbol{\alpha} = (\underbrace{\boldsymbol{H} \cdot \boldsymbol{W}_{Q,0}}_{\text{Original}} + \underbrace{(\boldsymbol{H} + \varepsilon)\boldsymbol{B}'\boldsymbol{A}'^{\top}}_{\text{Trainable}} + \underbrace{\varepsilon\boldsymbol{W}_{Q,0}}_{\text{Uncontrollable}})\frac{\boldsymbol{K}^{\top} \cdot \boldsymbol{V}}{\sqrt{d_k}},$$

The core challenge of robust fine-tuning lies in the conflict between the trainable and uncontrollable components. The uncontrollable term can introduce high-variance and sharp peaks into the attention distribution, causing it to fixate on irrelevant tokens. In essence, it must learn to **smooth** the erratic distribution induced by the perturbation to restore a stable, task-focused reasoning process.

prevailing approaches to robustness often operate *reactively*, focusing on aligning the final output of a perturbed input with that of a clean one, which forces the model to learn an internal correction implicitly. However, our analysis of the underlying mechanism motivates a *proactively* approach. We contend that a more principled method should not only regularise the final output (*an effect-driven "backward" view*) but also directly constrain the internal mechanism by compressing the attention shifts caused by input perturbations (*a cause-driven "forward" view*).

For Eq. 2:

$$L_{tr}^{\mathcal{D}} \leq \underbrace{L_{ex}^{\mathcal{D}_t}}_{\text{Empirical}} + \underbrace{\sqrt{\frac{D_{\text{KL}}(Q(\theta)\|P(\theta)) + \ln(\frac{2\sqrt{n}}{\delta})}{2n - 1}}}_{\text{Complexity}}, \quad \delta \in (0, 1)$$

The framework merges the probabilistic guarantees of PAC learning with the methodologies of Bayesian Inference, offering a data-dependent upper bound on the generalisation error through the distance between the posterior and prior beliefs of a model. For LLMs, another classical generalisation bound, the Vapnik-Chervonenkis Dimension, is not applicable due to their over-parametrised structures. Applying this framework to LLMs is made feasible by LoRA, which makes the data distribution's transformation over the hypothesis space computationally tractable. we seek to find an upper bound on the true risk $L_{tr}^{\mathcal{D}} := \mathbb{E}_{z\sim\mathcal{D}}L_{tr}(h_\theta, z)$ in terms of the empirical risk $L_{ex}^{\mathcal{D}_t} := \mathbb{E}_{z\sim\mathcal{D}_t}L_{ex}(h_\theta, z)$, given a confidence level of $1 - \delta$ and available training samples $z \sim \mathcal{D}_t$.

## C  LLM SEGMENT

In the main body of our work, we employ a computationally efficient punctuation-based method for segmenting model outputs. To validate this choice, we conducted an additional small-scale experiment using a more complex, high-cost segmentation approach powered by a pre-trained T5 model. This alternative method leverages the T5 model to perform semantic segmentation and alignment.

This experiment was conducted using the Bart-base model. The T5-based segmentation approach proved to be exceptionally resource-intensive, with a computational cost approximately 60 times higher than our standard punctuation-based method. Due to these practical constraints, we performed this validation on a smaller subset, using 1/8th of the original CNN/Dailymail and XSum datasets.

Fig. 4 below illustrates the learning trends of the inner-loop segment loss (the semantic shift loss $\mathcal{L}_{\text{sem}}$) for the first 600 fine-tuning steps (batch size: 32) on the CNN/DailyMail and XSum datasets, respectively.

### C.1  ANALYSIS OF RESULTS

From the comparison plots (Fig. 4), we can draw two key observations:

1. **General Convergence:** Both segmentation methods demonstrate a clear downward trend in segment loss on both datasets. This indicates that both the high-cost T5-based method and the efficient punctuation-based method are viable strategies, successfully guiding the

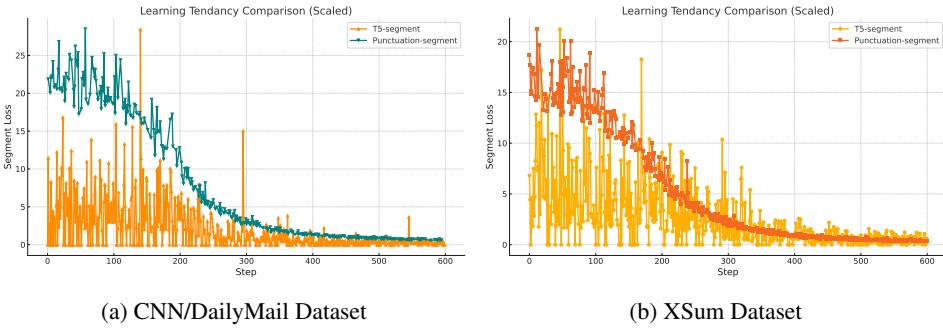

(a) CNN/DailyMail Dataset       (b) XSum Dataset

Figure 4: Comparison of segment loss learning trends for `bart-base` during the first 600 fine-tuning steps. The two sub-figures show the results on (a) a subset of the CNN/DailyMail dataset and (b) a subset of the XSum dataset. Both plots confirm the convergence of the segment loss for different segmentation methods.

    model to minimise the semantic discrepancy between outputs from clean and perturbed inputs.

2. **Training Dynamics and Sensitivity:** A notable difference emerges in the dynamic characteristics of the two loss curves. While both converge, the punctuation-based approach yields a smoother loss curve, whereas the signal from the T5-based segmentation is more volatile. We interpret this volatility not as training instability, but as an indicator of higher sensitivity. This characteristic likely stems from two aspects: first, the inherent complexity introduced by using a large pre-trained model as a segmentation tool; second, and more importantly, the finer granularity of the segmentation itself. By identifying more detailed semantic units, the T5-based method enables the loss function to more acutely capture the maximum distance between misaligned fragments. This heightened sensitivity to subtle semantic shifts—which are averaged out by the coarser punctuation-based method—directly supports our core hypothesis. It suggests that a more precise semantic segmentation reveals nuanced discrepancies, providing a more challenging but potentially more accurate optimisation signal, thus validating the importance of focusing on fine-grained semantic integrity.

# D  PSEUDOCODE FOR $S^2R^2$ FRAMEWORK

---

**Algorithm 1** Parameters Update Process of Semantic Segment Robustness Regularisation ($S^2R^2$)

---

**Notation:**

$L(\theta; x, y)$ denotes the supervised fine-tuning loss (e.g., Cross-Entropy). $\mathcal{L}_{\text{sem}}(\theta; x, x')$ denotes the semantic segment discrepancy from Eq. 3. $\mathcal{L}_{\text{att}}(\theta)$ denotes the LoRA-aware attention shift regulariser from Eq. 7. $\mathcal{P}(x)$ denotes the set of allowed perturbations for a clean input $x$.

**Require:**

Pre-trained LLM $f(\cdot; W_0, \theta)$; Dataset $\mathcal{X}$; Initial LoRA parameters $\theta_0$. Hyperparameters Learning rate $\alpha_{lr}$; Loss weights $\eta, \gamma$; Iterations $T$.

---

1: **for** $t = 1, \ldots, K$ **do**
2:   Sample $(x, y) \sim \mathcal{X}$.
3:   Inner-loop: Find worst-case perturbation via maximising a joint objective.
    $x'_k \leftarrow \underset{x' \in \mathcal{P}(x)}{\arg\max} \left\{ L(\theta_{k-1}; x', y) + \eta \cdot \mathcal{L}_{\text{sem}}(\theta_{k-1}; x, x') \right\}$
4:   Outer-loop: Update parameters via descending on the total $S^2R^2$ objective.
    $\mathcal{L}_{S^2R^2}(\theta_{k-1}) \leftarrow L(\theta_{k-1}; x, y) + \eta \cdot \mathcal{L}_{\text{sem}}(\theta_{k-1}; x, x'_k) + \gamma \cdot \mathcal{L}_{\text{att}}(\theta_{k-1})$
5:   $\theta_k \leftarrow \theta_{k-1} - \alpha_{lr} \cdot \nabla_\theta \mathcal{L}_{S^2R^2}(\theta_{k-1})$
6: **end for**
7: **return** $\theta_K$

---

# E    LORA ASSUMPTION VALIDATION

To validate our Assumption 2 regarding LoRA parameters as stated in the main paper, this section provides an analysis of the variance in their Frobenius norms over the course of a standard fine-tuning process (i.e., without our proposed $S^2R^2$). Assumption 2 posits that the Frobenius norms of the LoRA matrices, $||A^l||_F$ and $||B^l||_F$, are comparable.

Tab. 3 presents the aggregated Frobenius norm statistics for various models after fine-tuning on their respective datasets. The metrics are defined as follows:

- **A $F_{sum}$**: The sum of the Frobenius norms of matrix A across all LoRA layers, i.e., $\sum_l ||A^l||_F$.

- **B $F_{sum}$**: The sum of the Frobenius norms of matrix B across all LoRA layers, i.e., $\sum_l ||B^l||_F$.

- **LoRA $\Delta F_{sum}$**: The sum of the absolute differences between the norms of matrices A and B for each layer, i.e., $\sum_l |||A^l||_F - ||B^l||_F|$. This metric measures the symmetry or balance in the magnitudes of the LoRA matrices at each layer.

- **LoRA $ProdF_{sum}$**: The sum of the products of the norms of matrices A and B for each layer, i.e., $\sum_l (||A^l||_F \cdot ||B^l||_F)$.

| Model | Dataset | Method | A $F_{sum}$ | B $F_{sum}$ | LoRA $\Delta F_{sum}$ | LoRA $ProdF_{sum}$ |
|---|---|---|---|---|---|---|
| Bart-Base | CNN/DM | R | 19.275 | 9.784 | 57.515 | 184.472 |
| | | S | 18.542 | 7.831 | 64.595 | 140.354 |
| | Xsum | R | 17.241 | 9.103 | 49.177 | 154.976 |
| | | S | 15.977 | 6.562 | 57.687 | 100.997 |
| T5-base | PubMed | R | 49.139 | 18.435 | 208.239 | 875.407 |
| | | S | 40.838 | 18.282 | 142.747 | 691.655 |
| Mistral-7B | PubMed | R | 324.543 | 164.305 | 160.591 | 444.693 |
| | | S | 303.052 | 151.051 | 152.001 | 359.279 |

Table 3: Frobenius norm statistics of LoRA parameters after standard fine-tuning (without $S^2R^2$). Methods R and S correspond to the R3F and SMART baselines, respectively.

## E.1    ANALYSIS OF NORM COMPARABILITY

By examining the values of A $F_{sum}$ and B $F_{sum}$ in Tab. 3, we can empirically assess the validity of Assumption 2. The data reveals a consistent trend across all models, datasets, and baseline methods: while the norms of matrices A and B are not identical, they consistently remain within the same order of magnitude.

For instance, with the `Bart-Base` model, the ratio of A $F_{sum}$ to B $F_{sum}$ is approximately 2.0-2.4. For the larger `T5-base` and `Mistral-7B` models, this ratio remains in a similar range, approximately 2.0-2.7. In the context of neural network parameter magnitudes, a difference of a factor of 2-3 is generally considered comparable, especially when contrasted with scenarios where parameters might differ by several orders of magnitude. This observation indicates that neither matrix's norm grows disproportionately large while the other shrinks to near zero.

This empirical result aligns with the discussion in our main paper, which acknowledges the potential for asymmetry in the roles of matrices A and B while maintaining that their magnitudes are often observed to be comparable. Therefore, the data presented provides a solid empirical grounding for Assumption 2, justifying its use in the simplification of the KL divergence term within our PAC-Bayesian analysis.

# F Large Language Model Usage Statement

During the preparation of this manuscript, we utilised an LLM, specifically OpenAI's GPT-5, to assist with language editing and polishing. The primary uses of the LLM were for improving grammar, spelling, clarity, and overall readability.

We wish to clarify that all core scientific contributions, including the conceptualisation of ideas, the design of the methodology, the execution of experiments, and the interpretation of results, are entirely the work of the human authors. The LLM served exclusively as a writing aid and did not contribute to any of the substantive research aspects of this paper. The authors have carefully reviewed and edited all text generated or modified by the LLM and take full responsibility for the final content and its scientific accuracy.

