# OpenReview forum: "S²R²: Semantic Segment Robustness Regularisation on Prompt Perturbation"
_ICLR.cc/2026/Conference — ICLR 2026 Conference Withdrawn Submission_

### Official Review · Reviewer_b8ek · 2025-10-31

**Soundness:** 1
**Presentation:** 2
**Contribution:** 2
**Rating:** 2
**Confidence:** 5

**Summary:**

This paper proposes S²R², a new fine-tuning framework for enhancing the robustness of Large Language Models to prompt perturbations. The authors argue that existing robustness methods are often "holistic," failing to account for the non-uniform distribution of semantic importance across a sequence. Their approach operates on two core principles: (1) enforcing segment-level semantic alignment by splitting model outputs into semantic segments and minimizing the discrepancy between corresponding segments from clean and perturbed inputs using an Optimal Transport-based loss, and (2) penalizing perturbation-induced attention shifts through a LoRA-aware regularization term derived from a theoretical analysis of the attention mechanism. The authors connect their objective to the PAC-Bayesian framework, claiming that minimizing their proposed loss terms also tightens a certified generalization bound.

**Strengths:**

1.The paper makes a notable effort to ground the problem of LLM robustness in a formal theoretical framework. The connection drawn between the proposed regularization objective and the PAC-Bayesian generalization bound is a principled approach that attempts to move beyond purely empirical methods.

2.The authors conduct a thorough set of experiments across multiple model architectures  and datasets. The inclusion of various perturbation types and a zero-shot cross-dataset evaluation provides a broad assessment of the method's performance and transferability.

**Weaknesses:**

1.The central claims of the paper, upon critical deconstruction, appear to repackage well-established concepts with complex terminology. The central idea of "semantic asymmetry" that some parts of a text are more important than others is a foundational principle of NLP and is precisely what attention mechanisms were designed to address. The proposed "semantic segment" approach appears to be an attempt to explicitly enforce a mechanism that is already implicitly present. Moreover, the proposed "mechanism-oriented" attention shift loss, despite being justified through extensive mathematical derivations, ultimately simplifies to a penalty on the product of the Frobenius norms of the LoRA matrices. This is functionally very similar to standard weight decay, a long-established regularization technique, which raises questions about whether the complex theory is a post-hoc justification for a simple engineering practice.

2.The entire framework is highly dependent on the quality of its "semantic segmentation," yet this foundation appears weak. In the main experiments, the authors admit to using a simple heuristic based on "natural language punctuation" for efficiency. This severely undermines the claim that the method operates on deep semantic units, as splitting text by commas and periods is a syntactic, not semantic, operation. While Appendix C attempts to validate this with a more complex segmenter, the fact that a simple heuristic yields comparable results suggests that the sophisticated theoretical motivation may not be necessary, weakening the methodological contribution.

3.The experimental results reveal a critical flaw that is explained away with questionable post-hoc reasoning. In Table 1(c), the model trained with S²R² shows a Performance Drop Rate over 25 times worse than the SMART baseline on the PubMed dataset, indicating a catastrophic failure in robustness according to the standard ROUGE metric. The authors' defense that ROUGE is flawed and their method excels in "semantic self-consistency" is unconvincing. While ROUGE has limitations, a performance gap of this magnitude cannot be dismissed so easily and strongly suggests that the method may produce outputs that are dramatically divergent from the reference, a particularly concerning failure mode for a biomedical dataset.

**Questions:**

1.The paper's main experiments rely on a simple punctuation-based method for "semantic segmentation." How can the authors be certain that this method captures "deep semantic asymmetry" rather than just syntactic clauses? If a simple heuristic is sufficient, does this not diminish the claimed importance of operating on true "semantic" units?

2.How do the authors explain the 25-fold increase in PDR on the PubMed dataset compared to the SMART baseline? Rather than attributing this entirely to the flaws of the ROUGE metric, could this not indicate a failure mode where the model learns to generate outputs that are semantically consistent but factually or stylistically detached from the ground truth?

3.The evaluation is confined to the summarization task. How would the proposed segment-level alignment perform on tasks that are highly sensitive to exact token-level accuracy, such as question answering, code generation, or mathematical reasoning? In such tasks, a single incorrect keyword can render the entire output useless, a nuance that a coarse, segment-based L_sem might fail to capture.

---

### Official Review · Reviewer_VUjv · 2025-11-01

**Soundness:** 2
**Presentation:** 2
**Contribution:** 2
**Rating:** 4
**Confidence:** 2

**Summary:**

In this work, the prompt level perturbation robustness is studied and the $S^2R^2$ approach is introduced to improve the robustness of language model towards multiple types of perturbations: (1) Typographical & Deletion (2) Synonym Replacement and (3) Paraphrasing. Given a clean input and its perturbation, the objectives are : (1) Output-oriented segment-level semantic loss to penalize worst-case semantic shifts. (2) Mechanism-oriented attention shift loss constrains perturbation-induced changes in LoRA parameters. The proposed approach showed promising results in improving the robustness of language models.

**Strengths:**

- The proposed approach is easy to follow and understand.

**Weaknesses:**

- It's unknown that whether the adversarial training will impact the instruction following capability of language models. Usually the post-trained language model is able to follow the instructions in the prompt, which means the model is sensitive enough to response to the changes in the input prompt. It seems the robustness defined in this work conflict with the instruction following capability: when perturbation is introduced, the model should generate same output. It's understandable that the model should be robust to adversarial perturbation, but it's unknown that whether the model's "good" instruction following capability is affected or not.

**Questions:**

- It's unknown that whether the adversarial training will impact the instruction following capability of language models. Usually the post-trained language model is able to follow the instructions in the prompt, which means the model is sensitive enough to response to the changes in the input prompt. It seems the robustness defined in this work conflict with the instruction following capability: when perturbation is introduced, the model should generate same output. It's understandable that the model should be robust to adversarial perturbation, but it's unknown that whether the model's "good" instruction following capability is affected or not. Is it to possible to test on regular tasks to verify if there is regression of normal tasks?
- Besides the 3 perturbations tested, is it possible to test other perturbations?
- Is the proposed approach generalizable to other finetuning paradigms, like PEFT, RL and full finetuning?

---

### Official Review · Reviewer_L5Yr · 2025-11-04

**Soundness:** 3
**Presentation:** 3
**Contribution:** 2
**Rating:** 4
**Confidence:** 3

**Summary:**

This paper looks at the problem of prompt sensitivity in large language models—how small changes in prompt wording can produce very different answers. The authors propose a simple but structured way to make fine-tuning more robust, building on the standard LoRA (low-rank adaptation) framework.

Their idea is to regularize the LoRA updates in two ways. The first is an attention-shift loss, which penalizes how much the attention weights (modified through LoRA) change when the prompt is perturbed. The second is a semantic-shift loss, which measures the Optimal Transport (OT) distance between the embedding representations of outputs from the clean and perturbed prompts. The total training loss is a weighted sum of the task loss and these two regularizers.

The work is loosely motivated by a PAC-Bayesian generalization argument. The authors recall that the expected generalization gap can be bounded by the empirical loss plus a KL term between the fine-tuned (posterior) and pre-trained (prior) model distributions. They interpret the KL term as a robustness regularizer and relate it to the proposed attention-shift loss, although this connection remains qualitative. The semantic-shift loss, by contrast, is heuristic—it is introduced as a complement rather than a direct theoretical consequence.

Experiments are run on three summarization datasets (CNN/DailyMail, XSum, and PubMed) using BART, T5, and Mistral-7B. Perturbations include paraphrases and synonym substitutions. The results show modest but consistent improvements in ROUGE scores under perturbations, with little computational overhead. Ablation studies indicate that

**Strengths:**

1.	Timely topic. Robustness to prompt variations is an important and concrete problem for current LLMs.
	2.	Simple, practical method. The proposal integrates easily with LoRA and has negligible computational cost.
	3.	Clear writing and competent experiments. The empirical results are easy to follow and show that the method works as advertised.
	4.	Reasonable theoretical framing. The PAC-Bayesian interpretation gives some intuition for why constraining LoRA weights might help generalization.

**Weaknesses:**

1.	Superficial theory.
The PAC-Bayes argument is essentially borrowed from prior work and restated at a high level. The paper never computes a bound or verifies any PAC-Bayes quantity empirically. The attention-shift loss fits the general intuition of weight regularization, but the connection is informal, and the semantic-shift loss is entirely heuristic.
	2.	Limited novelty.
The method sits close to existing robust fine-tuning approaches such as SMART, R3F, and related adversarial-smoothing methods. All of these encourage local smoothness under perturbations, differing mainly in where the regularization is applied (output space, embedding space, or parameter space). Targeting LoRA-specific parameters is a practical twist but not a fundamentally new idea.
	3.	Narrow empirical scope.
All experiments involve summarization tasks. There are no results on QA, dialogue, or reasoning, though the paper claims general robustness for LLMs. The conclusions are therefore limited in scope.
	4.	Unclear evaluation details.
The perturbation process is not fully described (number of variations, randomness, or strength of perturbation). No confidence intervals or variance measures are reported. Code is promised but not provided in the anonymous version.
	5.	Ambiguity around the “semantic-shift” loss.
The OT component is not used in any meaningful transport sense—it’s effectively a pairwise embedding distance. The choice of metric is arbitrary, and it’s not clear how stable or necessary this term is.

**Questions:**

1.	Theory and connection to PAC-Bayes.
	•	Can you make the connection between the attention-shift loss and the PAC-Bayesian KL term explicit?
	•	Is there any empirical evidence that minimizing this loss reduces model divergence from the pre-trained prior?
	2.	On the semantic-shift loss.
	•	What cost metric and solver are used for the OT term? Is an actual transport plan computed or is this effectively a distance between mean embeddings?
	•	How sensitive are the results to this choice, and why is this formulation preferable to a simpler cosine-similarity penalty?
	3.	Relation to prior work.
	•	How does your method differ concretely from SMART or R3F, beyond operating in the LoRA parameter subspace?
	•	Why not include those baselines in your comparisons?
	4.	Evaluation scope.
	•	Have you tried this approach on QA or dialogue tasks, where robustness is arguably more challenging?
	•	Are improvements consistent across different types of perturbations—lexical, syntactic, or semantic?
	5.	Interpretation.
	•	What exactly do the regularization terms do to the learned LoRA weights? Do they make updates smaller, sparser, or more correlated across prompts? Some analysis of the learned parameters would make the story more convincing.

---

### Official Review · Reviewer_yEMG · 2025-11-06

**Soundness:** 2
**Presentation:** 3
**Contribution:** 2
**Rating:** 2
**Confidence:** 4

**Summary:**

This paper proposes S²R², a fine-tuning framework aimed at improving LLM robustness against prompt perturbations. The method introduces two complementary objectives: a semantic segment loss that penalizes meaning shifts in key output segments, and an attention shift loss that regularizes changes in cross-attention patterns. The core idea is to move beyond holistic output alignment and focus on semantically important segments. The method is implemented via LoRA-based parameter-efficient fine-tuning and theoretically grounded using a PAC-Bayesian generalization bound. Experiments on summarization datasets demonstrate improved robustness and tighter generalization guarantees compared to existing baselines.

**Strengths:**

1. The paper introduces a segment-level robustness loss that explicitly targets semantic drift in critical parts of the output.

2. The attention shift regularizer is theoretically well-motivated and connected to the PAC-Bayesian framework.

3. The experiments cover multiple model architectures and domains, which validates its effectiveness.

**Weaknesses:**

1. The key assumption that robustness issues are primarily tied to “key semantics” or a few important keywords may be questionable. In practice, even seemingly non-critical tokens—such as punctuation marks or meaningless characters—can significantly affect model behavior.

2. The discussion of related work is limited to brief mentions in the Introduction and Preliminaries, and is far from sufficient. There exists a substantial body of prior research on LLM robustness at various levels of granularity—not just at the sequence level—including both token-level [1, 2, 3] and segment-level [4, 5] methods. The paper appears to overlook this line of work and does not adequately differentiate itself from the existing literature. In particular, the discussion of robust training methods is overly simplistic and lacks depth.

3. The use of LoRA is not clearly motivated. It remains unclear why the proposed method must be implemented in a LoRA setting rather than within a standard fine-tuning framework. A stronger justification or motivation for this design choice would strengthen the contribution.

References

[1] On Subword Robustness in Large Language Models

[2] ERICT: Enhancing Robustness by Identifying Concept Tokens in Zero-Shot Vision Language Models

[3] Interpreting token compositionality in LLMs: A robustness analysis

[4] PEARL: Towards Permutation-Resilient LLMs

[5] Enhancing Noise Robustness of Retrieval-Augmented Language Models with Adaptive Adversarial Training

**Questions:**

Please refer to the above part.

---

### Note · Authors · 2025-11-28

I have read and agree with the venue's withdrawal policy on behalf of myself and my co-authors.